# Hormonal Contraception and the Risk of Breast Cancer in Women of Reproductive Age: A Meta-Analysis

**DOI:** 10.3390/cancers15235624

**Published:** 2023-11-28

**Authors:** Luz Angela Torres-de la Roche, Angélica Acevedo-Mesa, Ingrid Lizeth Lizarazo, Rajesh Devassy, Sven Becker, Harald Krentel, Rudy Leon De Wilde

**Affiliations:** 1University Hospital for Gynecology, Pius Hospital, University Medicine Oldenburg, Carl von Ossietzky University, 26121 Oldenburg, Germany; maacevedom@unal.edu.co (A.A.-M.); illizarazor@unal.edu.co (I.L.L.); secretariat@drrajeshdevassy.com (R.D.); rudy-leon.dewilde@pius-hospital.de (R.L.D.W.); 2University Hospital for Gynecology and Obstetrics, University Hospital Frankfurt, 60596 Frankfurt, Germany; sven.becker@kgu.de; 3Clinic of Gynecology, Obstetrics, Oncology, and Senology, Bethesda Hospital, 47053 Duisburg, Germany; krentel@cegpa.org

**Keywords:** breast cancer, hormonal contraception, risk factors, premenopausal, meta-analysis

## Abstract

**Simple Summary:**

Breast cancer (BC) is caused by highly invasive and metastatic malignant tumors and affects a large number of women, threatening their health and quality of life. Previous studies have identified mixed evidence on the relationship between the use of hormonal contraceptives (HC) and the development and progression of breast cancer, especially in women of reproductive age. Our results suggest that there is a higher risk of BC in ever-users of HC, but attempting to determine the risk of developing BC among premenopausal women is difficult due to the high heterogeneity observed among the studies, indicating that our results should be approached with caution when translating them into practice.

**Abstract:**

This study aims to summarize evidence from observational studies about the lifetime use of HC and the risk of BC in women of reproductive age. The PubMed, Cochrane, and EMBASE databases were searched for observational studies published from 2015 to February 2022. Meta-analyses were performed using adjusted odds ratios and relative risks with a random-effects model using the I^2^ statistic to quantify the heterogeneity among studies. Of the 724 studies identified, 650 were screened for title/abstract selection, 60 were selected for full-text revision, and 22 were included in the meta-analysis. Of these, 19 were case-control studies and 3 were cohort studies. The results of the meta-analysis indicate a significantly higher risk of developing BC in ever users of HC (pooled OR = 1.33; 95% CI = 1.19 to 1.49). This effect is larger in the subgroups of case-control studies (pooled OR = 1.44, 95% CI = 1.21 to 1.70) and in the subgroup of studies that strictly define menopausal status (pooled OR = 1.48; 95% CI, 1.10 to 2.00). Although our meta-analysis of observational studies (cohort and case-control) suggests a significantly increased overall risk of BC in users or ever-users of modern hormonal contraceptives, the high heterogeneity among studies (>70%) related to differences in study design, measurement of variables, confounders, among other factors, as well as publication biases should be considered when interpreting our results.

## 1. Introduction

Breast cancer (BC) is caused by highly invasive and metastatic malignant tumors and affects a large number of women, threatening their health and quality of life. Given that the development of breast cancer responds to a combination of several biological, psychological, and environmental factors, including the intake of exogenous hormones, the causal mechanism cannot be directly linked to only one risk factor [1]. Additionally, female hormones play an important role during the development of sexual characteristics in the breast glands, acting as mitogens able to promote the progression of a wide range of cancers through a variety of receptor-dependent signaling pathways [2]. Molecular mechanisms change over time because the mammary gland’s sensitivity to each hormone varies with women’s developmental stages [3]. During normal development, estrogen (E) signaling is required for pubertal ductal elongation, while progesterone’s (P) actions on mammary epithelial cells (MEC) are required for ductal side branching and alveologenesis. Prolactin (PRL) is important for epithelial cell proliferation and milk production in the mammary gland [3]. It has autocrine and paracrine functions and exerts its biological activities through membrane PRL receptors (PRLr), which are members of the cytokine receptor superfamily. The PRL and PRLr pathways are important regulators in mammary gland development through PRL generation [3] and PRLr gene knockout [4]. PRLr signaling drives further mammary alveologenesis during pregnancy and is required for milk production and secretion. Furthermore, E and P are expressed in different normal MEC populations, with extensive paracrine crosstalk between E and P signaling as well as between P and PRL signaling, resulting in a continuum of overlapping and highly integrated signaling pathways that are critical for genomic integrity and cancer susceptibility. Exogenous hormones, like those contained in hormonal contraceptive drugs, could affect different signaling pathways, increasing the risk of BC [2,5]. 

Studies on the mechanisms of mutagenesis report that E and P act on the P-450 1BI enzyme complex and are found in abundance in breast and other tissues, hydrolyzing estrogens to produce catechol reactive estrogens or to produce reactive oxygen species [6]. Other genome-sequencing analyses have demonstrated genome-wide DNA mutational signatures related to cytidine deamination by APOBEC3 genes (A3A and A3B), a family of enzymes that mutate RNA or DNA by deaminating cytidine to uridine [7]. These, in turn, cause DNA strand breaks that disrupt DNA repair mechanisms and contribute to the genomic instability that drives cancer development, as seen in BRCA1 and BRCA2 mutations [5,6,7]. 

Regarding synthetic estrogens and progestins used for contraception or hormone replacement therapy during menopause, epidemiological studies and clinical trials show conflicting results regarding the risk of developing BC. These inconsistencies could be the result of multiple factors such as the design of the studies, the confounding factors considered in each study, the non-comparability of the data from the reviews, the possibility of an undetected malignancy at the time of initiation of hormonal contraception (HC), or the heterogeneity of the biological effect of steroid hormones on breast cells in the presence of other risk factors for BC. According to different meta-analyses of case-control studies, the use of modern oral contraceptives has different effects on the risk of BC subtypes. In 2017, Li et al. [8] reported a statistically significant higher risk of triple-negative breast cancer (TNBC) among users (OR = 1.31, 95% CI = 1.18–1.45) in comparison to a healthy population (OR = 1.21, 95% CI = 1.01–1.46). Similar results were found in two meta-analyses published recently. Barańska et al. [9] reported an increased risk of TNBC (OR = 1.37; 95% CI: 1.13 to 1.67) and negative-estrogen receptor (ER-) cancers (OR = 1.20; 95% CI: 1.03 to 1.40), but reduced risk for positive-estrogen receptor (ER+) tumors (OR = 0.92; 95% CI: 0.86 to 0.99). Mao X et al. [10] showed that prolonged oral contraceptive use is associated with a 16% higher risk of TNBC (RR: 1.16; 95% CI: 1.05 to 1.29), but found no statistically significant associations for luminal A, luminal B, or HER2 tumors. The evidence is scant since just a few prospective controlled studies have been conducted in the past decade. 

Modern low-dose HC methods, which are available in a variety of hormonal combinations and forms of administration (oral, injectable, transdermal, subdermal, intrauterine, and intravaginal), have proved to be effective in preventing pregnancy in healthy, young, and premenopausal women. However, the reported risk of BC development is still one of the most frequent reasons given for the non-acceptance of HC use [11], even though the long-term benefits of HC are thought to be multiple and greater than this potential risk [12,13].

Given the aforementioned background, the main aim of this study is to investigate the risk of lifetime use of HC in the development of BC in women of reproductive age. To this end, we aim to summarize the existing evidence regarding the relationship between HC use and BC as reported in peer-reviewed observational studies conducted with women of reproductive age by performing a systematic review of the literature with a meta-analysis. In addition, we aim to investigate whether differences in study design, report of menopausal status, and quality of the studies influenced the heterogeneity of the results. 

## 2. Materials and Methods

This systematic review and meta-analysis adhere to the guidelines of the Preferred Reporting Items for Systematic Reviews and Meta-Analyses (PRISMA) [14]. Additionally, we followed the guidelines for Meta-Analysis of Observational Studies in Epidemiology (MOOSE) [15], given that one of our selection criteria for studies was that they were observational. The protocol for this meta-analysis was registered in the International Prospective Register of Systematic Reviews (PROSPERO) [16] with registry ID, CRD42022304693.

### 2.1. Identification of Eligible Studies 

Our research question and eligibility criteria were specified using the PIICOS framework from the Centre for Evidence-Based Medicine, Oxford University [17], as shown below.

### 2.2. Participants 

Eligible studies included adult participants of the female sex, within the range of reproductive age, defined by the researchers as ranging from 18 to 55 years old. We included studies in which participants had used hormonal contraception at any time and for any duration within their reproductive age range. Given that the different health effects of hormonal contraceptives could be present when using them at earlier ages, we excluded studies performed with underaged women. 

### 2.3. Exposure 

We included studies in which the exposure variable was the use of hormonal contraceptives, including six main types of hormonal contraception: oral contraceptives, injectable contraceptives, intravaginal devices, intrauterine devices, transdermal patches, and subdermal implants. We classified the exposure based on administration modality and not on the hormonal type of the contraceptives, given that we aim to explore differences in breast cancer risk based on the administration route of contraceptives and because most studies did not report the name of the pharmaceutical formulation of the contraceptive. We considered exposure of any duration or frequency. 

### 2.4. Outcome 

The risk of breast cancer in women of reproductive age was defined as our outcome measure. We included studies that assessed the presence of breast cancer through independent validation (e.g., diagnosis made by a medical professional), record linkage, or self-reports.

### 2.5. Type of Studies 

We included original observational studies, including case-control studies, and prospective and retrospective (historic cohorts) cohort studies. We excluded cross-sectional studies and case-only studies. We only considered peer-reviewed studies. Included studies had to provide a clear measure of the exposure and the outcome, and the risk estimates of the exposure or enough information to calculate them.

### 2.6. Search Strategy 

On the 8th February 2022, the databases of PubMed, Cochrane, and EMBASE were consulted to identify observational studies including prospective cohort studies, case-control studies, and historic cohort studies. We developed a research strategy for each database (see Appendix A) based on three main sets of terms including the following: (a) terms related to the exposure variable, namely, hormonal contraception of any type; (b) terms related to breast cancer; and (c) terms related to risk. We used the Boolean operator OR to combine terms within each set and the Boolean operator AND to combine the three sets. The search strategy was first developed for PubMed and then adapted for Cochrane and EMBASE. In the strategy search for PubMed, we included an additional string to restrict the search to observational studies. All searches were restricted to articles from 2015 until February 2022, given that we aim to focus on updated results from modern hormonal contraceptives. In addition, a manual search of other sources on Google Scholar was performed with the term “hormonal contraceptive use and breast cancer risk”, as was a manual cross-reference search based on the reference list of the studies found.

### 2.7. Data Selection and Extraction 

All articles retrieved from each database and manual search were downloaded to an open-sourced software for reference management, Zotero© version 6 [18]. For counting purposes, articles were organized in a folder corresponding to each database or manual search. Afterward, all the articles were stored in one folder to identify and remove duplicates. For the title/abstract selection process, all titles and abstracts of the retrieved articles were compiled in an Excel workbook with instructions for the fulfillment of the eligibility criteria. Two authors with expertise in obstetrics, gynecology, and cancer research independently screened the titles and abstracts, answering the following questions with “yes” or “no”, to determine eligibility criteria: (a) is it an observational study? (b) is the population adult women of reproductive age? (c) is there exposure to a hormonal contraceptive agent (measured by a clinical record or self-report)? (d) what is the outcome of breast cancer (measured by medical diagnosis, clinical record, or self-report)? (e) were quantitative data reported? (e.g., odds ratios, risk ratios, hazard ratios). Any disagreement was resolved by consensus between the two researchers. If an agreement was not achieved, a third author was consulted.

All articles selected for full-text revision were retrieved and stored as a project in the Zotero web cloud. The Systematic Review Data Repository (SRDR+) from the Agency for Healthcare Research and Quality [19] was used to extract the data from the full-text revisions. A methodologist designed an extraction form using the extraction form builder of SRDR+, and this included the key data to extract from each article. SRDR+ automatically extracted the full references, authors’ names, titles, abstracts, and years of publication of each article. Two independent authors extracted the following available data—(a) study design: aim, type of study, and follow-up time if cohort study; (b) participants: population description, sample sizes, and selection criteria; (c) outcome: definition, assessment, and cancer type if available; (d) exposure: definition, assessment, hormonal contraceptive type by administration route, and type of contraceptive hormone; (e) risk estimates and quantitative information for the calculation of risk estimates: sample sizes, number of breast cancer cases, number of exposed cases and controls, number of unexposed cases and controls, type of statistical analysis used, and crude risk estimates (i.e., odds ratios, relative risks, adjusted risk estimates, confidence intervals and *p*-values of risk estimates, covariates used for adjusted analysis, and duration of use). 

### 2.8. Quality Assessment 

For the quality assessment, we employed the Newcastle-Ottawa Quality Assessment Scale (NOS) for cohort and case-control studies provided by the SRDR+ [20,21]. This scale contains an assessment of the selection, comparability, and outcome in cohort studies. For case-control studies, the NOS assesses the selection, comparability, and exposure. Every study can obtain a maximum of 8 points. For this meta-analysis, based on previous literature, studies were classified as having very high risk of bias/low quality (0 to 3 NOS points), high risk of bias/moderate quality (4 to 6), and low risk of bias/high quality (7 to 8) [22]. Two independent researchers assessed the quality of the articles included in the systematic review when performing the data extraction using the SRDR+ extraction form. Any disagreement was solved by consensus or by a third researcher.

### 2.9. Statistical Analysis 

To assess interrater reliability in the title and abstract selection, a researcher with expertise in epidemiology and methodology calculated the percentage of agreement and the Kappa coefficient. Kappa values between 0.41 and 0.60 indicate moderate agreement, 0.61 and 0.80 indicate substantial agreement, and 0.81 and 1.00 indicate almost perfect agreement [23].

Given that our outcome measure was binary, risk estimates such as odds ratios (ORs) and relative risks (RRs) were extracted for the meta-analysis. RRs were converted into ORs for consistency when pooling the estimates using the software RevMan Version 5.0 [24]. When articles reported estimates according to pre- and post-menopausal status, only the estimates reported for pre-menopausal women were extracted. To perform the meta-analysis, first, the crude ORs of the studies were recalculated, and thereafter the pooled OR and 95% CI were calculated with RevMan.5 using a random-effects model. The heterogeneity among studies was explored using the Chi^2^ test, and the statistic I^2^ was used to quantify the heterogeneity between studies. The risk of publication bias was visually explored with a funnel plot. 

### 2.10. Subgroup Analyses 

Additional analyses were performed to explore sources of heterogeneity, identify differences between subgroups, and explore potential differences in the pooled estimates of the studies included in the present analysis. Pooled OR and 95% CI for each subgroup were calculated using a random-effects model, and significant differences between subgroups were explored; likewise, the I^2^ of each group was quantified to observe changes in heterogeneity.

Firstly, a subgroup analysis based on the study design, comparing prospective cohort studies against case-control studies, was performed using RevMan Version 5.0. Secondly, a subgroup analysis based on the report of menopausal status was performed to explore potential differences in the pooled BC risk estimates for pre-and postmenopausal women. Studies that reported estimates for pre-menopausal women were compared to studies that did not report such a distinction. It is important to note that this meta-analysis aimed to include all studies referring to women of reproductive age (18 to 55). However, some studies presented estimates according to menopausal status, whereas others reported one estimate for the whole group. Thirdly, a sensitivity analysis was performed to explore whether heterogeneity was related to study quality. Considering the NOS score, studies with high to moderate risk of bias and low to moderate quality were compared against studies with low risk of bias and high quality—that is, studies with a score of 7 to 8 were compared to studies with a score lower than 7. For this analysis, we compared only case-control studies given that they share the same scale for quality assessment. 

## 3. Results

The search strategy identified 724 studies for possible evaluation (Figure 1), most of which were from the PubMed and Embase repositories. Once the duplicates were removed, 89.78% (*n* = 650) were screened for title/abstract selection. After title/abstract selection, the independent reviewers had an 86% agreement, indicating a moderate to high level of agreement. The Kappa statistic is 0.72, which also indicates a moderate level of agreement between the raters. These results suggest that the raters have a moderate to high agreement on the articles selected for full-text data extraction. From the original number of articles retrieved, 8.29% (*n* = 60) were left for full-text evaluation. Finally, 22 (17.74%) studies were included in the quantitative analysis.

This meta-analysis included data from 22 reports showing the results of 19 case-control studies, 2 prospective cohort studies, and 1 cross-sectional cohort study that involved a total of 2,275,304 women and from which 44,295 BC cases (1.95%) were reported (Table 1). The two prospective cohort studies included had a mean follow-up of 8.95 years.

Most of the studies were conducted in Western Asia (*n* = 8, 36.36%) [25,26,27,28,29,30,31,32], followed by five in North Europe [33,34,35,36,37], three in East Asia [38,39,40], two in África [41,42], two in the USA [43,44], and two in Indonesia [45,46] but none in Latin America. However, 98% of the population studied was included in the European studies, with the largest population being included in the Danish prospective cohort study that involved 1,797,932 women [35]. The smallest studies were retrospective case-control analyses from Iran [25] and Jordan [29] involving 450 women each.

**Table 1 cancers-15-05624-t001:** Characteristics of studies included in the meta-analysis of hormonal contraceptive use as a potential risk factor for breast cancer.

Authors, Year of Publication, Country	Title	Study Aim/Research Question	Type of Contraceptive	Research Design and Methods (Follow-Up Time for Cohort Studies)	Sample
Heikkinen et al. (2016), Finland [33]	Use of exogenous hormones and the risk of breast cancer: results from self-reported survey data with validity assessment.	To estimate the association between the use of exogenous hormones and BC risk and to assess the validity and representativeness of the Women’s Health and Use of Hormones (WHH) survey concerning various background and lifestyle factors.	Oral and Intrauterine device	Prospective Cohort 7 years follow-up, 2000–2007	*n* = 25,560 5927 cases 19,633 controls
Ichida et al. (2015), Japan [38]	No Increase in Breast Cancer Risk in Japanese Women Taking Oral Contraceptives: A Case-Control Study Investigating Reproductive, Menstrual and Familial Risk Factors for Breast Cancer.	To investigate OC use and breast cancer risk, as well as menstrual, reproductive, and family factors.	Oral	Case-control	*n* = 12,378 155 cases: (106 premenopausal women/119 ever users/36 ever users) 12,333 Controls (10,427 premenopausal women/8698 ever users/3525 ever users)
Joukar et al. (2016), Iran [25]	The Investigation of Risk Factors Impacting Breast Cancer in Guilan Province.	To determine the factors influencing breast cancer in women referred to health centers in Guilan province in 2015–2016.	Oral	Case-control	*n* = 450 225 cases (102 premenopausal women; 90 ever users, 132 never users) 225 controls (110 premenopausal women; 84 ever users, 141 never users)
Nguyen et al. (2016), Vietnam [39]	A Matched Case-Control Study of Risk Factors for Breast Cancer Risk in Vietnam.	To identify hormonal, reproductive, and anthropometric risk factors for both pre and postmenopausal breast cancer in Vietnamese women.	Oral	Case-Control	*n* = 1798 492 cases (196 premenopausal) 1306 control (196 premenopausal)
Balekouzou et al. (2017), Central African Republic (Bangui) [41]	Reproductive risk factors associated with breast cancer in women in Bangui: a case-control study.	To determine the relationship between breast Ca and reproductive factors in women living in Bangui.	No type specified	Case-control 1:2 age-matched	*n* = 522 174 cases 348 controls
Mørch et al. (2017), Demark [35]	Contemporary Hormonal Contraception and the Risk of Breast Cancer.	To assess the associations between the use of hormonal contraception and the risk of invasive breast cancer.	Oral, Intrauterine device, Intravaginal device, Transdermal patch, Subdermal implant	Prospective Cohort 10.9 years follow-up	*n* = 1,797,932 11,517 cases
Dianatinasab et al. (2017), Iran [26]	Hair Coloring, Stress, and Smoking Increase the Risk of Breast Cancer: A Case-Control Study.	To evaluate a wide range of potential risk factors of BC in a representative sample of Iranian women.	Oral	Case-control	*n* = 1052 526 cases (309 premenopausal; 280 never-users, 246 ever-users 526 controls (321 premenopausal women; 312 never-users, 214 ever-users
Jareid et al. (2018), Norway [34]	Levonorgestrel-releasing intrauterine system use is associated with a 1 decreased risk of ovarian and endometrial cancer, without an increased risk of breast cancer. Results from the NOWAC Study.	To combine self-reported information on OC use and reproductive factors from the Norwegian Women and Cancer (NOWAC) Study.	Oral and Intrauterine device	Prospective Cohort Study 12.5 years follow-up	*n* = 104,318 297 cases (9144 ever users, 95,174 never users)
Chaveepojnkamjorn et al. (2017), Thailand [45]	Relationship between Breast Cancer and Oral Contraceptive Use among Thai Premenopausal Women: a Case-Control Study.	To determine the associations of BC with oral contraceptive (OC) use among Thai premenopausal women (TPW).	Oral	Case-control	*n* = 514 257 cases 257 controls
Chollet-Hinton et al. (2017), USA [43]	Biology and Etiology of Young-Onset Breast Cancers among Premenopausal African American Women: Results from the AMBER Consortium.	To examine tumor characteristics and breast cancer risk factors associated with premenopausal young (<40) vs. older (>40).	Oral	Case-control	*n* = 1775 354 cases 1167 controls
Wahidin et al. (2018), Indonesia [46]	Oral Contraceptive and Breast Cancer Risks: a Case-Control Study in Six Referral Hospitals in Indonesia.	Oral Contraceptive and Breast Cancer Risks: a Case-Control Study in Six Referral Hospitals in Indonesia.	Oral	Case-control	*n*= 762 381 cases 381 controls
Al-Ajmi et al. (2018), United Kingdom [36]	Risk of breast cancer in the UK biobank female cohort and its relationship to anthropometric and reproductive factors.	To explore the relationships of risk factors and breast cancer in the UK Biobank initiative.	Oral	Case-control	*n* = 57,707 618 cases (565 premenopausal ever users) 57,089 controls (50,012 premenopausal ever users)
Brinton et al., (2018), USA [44]	Breast cancer risk among women under 55 years of age by joint effects of usage of oral contraceptives and hormone replacement therapy.	To assess the effects on breast cancer risk of exposure to both oral contraceptives and menopausal Hormones.	Oral	Case-control	*n* = 1454 783 cases 671 controls
Yuan et al. (2019), China [40]	Induced Abortion (IA), Birth Control Methods, and Breast Cancer Risk: A Case-Control Study in China.	To explore the effect of common birth control methods and IA on breast cancer in Chinese women.	Oral	Case-control	*n* = 1599 794 cases 805 controls.
Alipour et al. (2019), Iran [27]	A Case-Control Study of Breast Cancer in Northeast of Iran: The Golestan Cohort Study.	To determine risk factors for BC and estimate the overall survival rate in BC patients of the Golestan Cohort Study (GCS)	Oral	Case-control	*n* = 499 99 cases 400 controls
Moradinazar et al. (2019), Iran [28]	Hormone therapy and factors affecting fertility of women under 50 years old with breast cancer.	to investigate the effect of factors related to fertility and hormone use on the risk of breast cancer in women aged under 50 years old in the west of Iran.	Oral, Injection, Intrauterine device, Subdermal implant	Case-control	*n* = 620 212 cases (ever users 151/never users 61) 408 controls (ever users 265/never users 43)
Bardaweel et al. (2019), Jordan [29]	Oral contraceptive and breast cancer: do benefits outweigh the risks? A case-control study from Jordan.	To explore any possible correlation between the contemporary and duration of OC use among Jordanian women and the risk of breast cancer.	Oral	Case-control	*n* = 450 225 cases 225 controls
Hamdi-Cherif et al. (2020), Algeria [42]	Sociodemographic and Reproductive Risk Factors for Breast Cancer: A Case-Control Study in the Setif Province, Northern Algeria.	To investigate the role of sociodemographic characteristics and reproductive factors in the etiology of BC in this young Arab/Berber population of Setif.	Oral	Case-control	*n* = 1227 612 cases 615 controls
Almasi-Hashiani et al. (2021), Iran [30]	The causal effect and impact of reproductive factors on breast cancer using super learner and targeted maximum likelihood estimation: a case-control study in Fars Province, Iran.	To estimate the causal effect of reproductive factors on BC risk in a case-control study using the double robust approach of targeted maximum likelihood estimation.	Oral	Case-control	*n* = 1715 787 cases 928 controls
Motie et al. (2021), Iran [31]	Breast cancer risk factors: A case-control study in Iranian women.	To identify the risk factors of breast cancer among Iranian women in the Khorasan province.	Oral	Case-control	*n* = 460 230 cases (119 ever users) 230 controls (49 ever use)
El Sharif et al. (2021), Palestine [32]	Reproductive Factors and Breast Cancer Risk in Palestine: A Case-Control Study.	To investigate the reproductive determinants of breast cancer among women in the West Bank of Palestine.	Oral	Case-control	*n* = 474 237 cases 237 Controls
Karlsson et al. (2021), United Kingdom [37]	Time-Dependent Effects of Oral Contraceptive Use on Breast, Ovarian, and Endometrial Cancers.	To clarify the time-dependent effects of long-term oral contraceptive use and cancer risk.	Oral	Cross-sectional Cohort with pro and retrospective design. Between 2006 and 2010	*n* = 256,661 17,739 BC cases (13,937 ever users, 3773 never users)

BC = Breast cancer; BMI = Body mass index; HR IUD = Hormone-releasing intrauterine device; HRT = Hormone replacement therapy; IA = Induced abortion; OC = Oral contraceptive pill.

### 3.1. Quality of Studies 

Regarding the quality criteria (Figure 2 and Appendix B), all studies were found to have well-defined and representative cases and controls; histologically confirmed breast cancer cases and unconfirmed cases were excluded from the analysis. However, the method of exposure ascertainment was different and of low quality in around 75% of the studies, as most of the studies used self-reported questionnaires or unblinded interviews, the hormone formulation of the contraceptive was not discriminated, the exact time of HC use was not based on records, or the missing data management of non-respondents was not stated. Only three case-control studies (13.6%% of the 22 included studies) [25,36,44] used tools to avoid recall bias, and two of the cohort studies (9.09% of the 22 included studies) [35,37] used accurate instruments to collect clinical information.

### 3.2. Exploration of Publication Bias 

Appendix C shows a funnel plot depicting the 22 included studies in the meta-analysis. As shown, the plot does not depict a symmetrical disposition. Most of the studies are located in the superior part of the graph, suggesting that the studies included show a relatively low standard error (SE log|OR| < 0.4). This could be due to most of the studies having large sample sizes, which means that we may not have captured smaller studies. We may have also captured more studies reporting significant associations, whereas studies reporting insignificant or negative associations may have not been included.

The main difference observed between the studies was the covariates used to adjust the statistical analysis. (Table 2). In addition to age and history of HC consumption, these factors were related to demographic, geographic, family, habit, gynecological, and economic aspects of participants. The reasons given for the choice of multiple factors for group adjustment were related to previous evidence of the best-known risk factors for the development of BC. Therefore, it was possible to analyze the relationship between exposure to HC and the risk of developing BC. 

### 3.3. Overall Risk Analysis

When including the 22 selected studies in the meta-analysis, the pooled OR is significant (*p* < 0.01%), which indicates a 33% (pooled OR = 1.33; 95%CI = 1.19 to 1.49) higher risk of breast cancer in ever users of hormonal contraceptives compared to non-users of hormonal contraceptives (Figure 3). However, it is important to note that there is substantial heterogeneity in the studies included, as indicated by a significant $I^{2}$ value of 82%. This suggests that the studies may have had important differences in terms of design, population, and methodology among other factors that could influence the outcomes. Despite the heterogeneity, the pooled OR suggests a moderate effect size, indicating that the use of hormonal contraceptives increases the odds of breast cancer risk by approximately 33%.

### 3.4. Subgroup Analysis

To explore the source of heterogeneity among the included studies, we performed several subgroup analyses to better understand the factors that could impact the reasons behind the heterogeneity and potential differences in the estimates of subgroups of studies.

**Subgroup analysis based on distinction of menopausal status:** Although the criteria for inclusion aimed to include studies on premenopausal women, only 7 of the 22 studies included in this meta-analysis made a clear distinction among participants according to menopausal status (Figure 4). Therefore, we analyzed the differences between pooled ORs and compared the heterogeneity between the group of studies that distinguished by pre-menopausal status (7 studies) and the group of studies that did not make such a distinction (15 studies).

The studies in which the ORs of only pre-menopausal women were distinguished showed a slightly lower heterogeneity (I^2^ = 79%) and a significantly (*p* < 0.01) higher risk of breast cancer (pooled OR = 1.48; 95% CI, 1.10 to 2.00). When observing the results from the studies that did not distinguish between pre-menopausal and post-menopausal participants, the heterogeneity is slightly higher (I^2^ = 82%) and the risk of breast cancer appears to be lower (OR = 1.33; 95% CI, 1.12 to 1.44).

This suggests that when looking at studies that strictly define menopausal status, the heterogeneity slightly improves. Moreover, when using data on strictly defined pre-menopausal women, the risk of breast cancer appears to be higher (48%) in women who used hormonal contraception versus women with no lifetime use of hormonal contraception. However, there are no significant differences between the subgroups (*p* = 0.34), which suggests that the menopausal status definition may not have had a great impact on the heterogeneity of the studies.

**Subgroup analysis based on the distinction of study design**: From the 22 studies included in the meta-analysis, 3 were cohort studies and 19 were case-control studies (Table 2). According to the type of study (Figure 5), the pooled analysis from the cohort studies did not show a significant increase in breast cancer risk in HC users (pooled OR 1.10; 95% IC, 0.95 to 1.28; *p* = 0.2). However, the study by Mørch et al. [35] showed a 23% increase in the risk of BC (OR = 1.23; 95% CI, 1.16 to 1.30), which could be explained by the large sample size (*n* = 11,517 cases). Contrary to this, the results from the studies by Jareid et al. [34] (*n* = 297 cases) and Karlsson et al. [37] (*n* = 17,056 cases) were not significant. Moreover, within this subgroup, there was very high heterogeneity (I^2^ = 92%). Therefore, an additional analysis was performed according to the quality of the studies.

It was found that the heterogeneity of the group of higher quality studies was slightly higher (I^2^ = 74%) than that of the group of lower quality studies (I^2^ = 68%), but there were no significant subgroup differences (*p* = 0.40). This suggests that the slight differences in quality may not have had a great influence on the heterogeneity estimate of the overall meta-analysis. 

Regarding the group of case-control studies, the pooled analysis showed a significant association (*p* = < 0.01) between HC use and breast cancer risk, with 44% increased odds (pooled OR = 1.44, 95% CI = 1.21 to 1.70). Specifically, of all the case-control studies, 10 reported a significantly increased risk of BC risk in lifetime users, 8 reported non-significant associations, and only Balekouzou et al. [41] reported a non-significant decreased risk of BC in a sample of African women users (OR = 0.62; 95% CI, 0.41 to 0.94).

## 4. Discussion

Contraceptive methods are known to confer important non-contraceptive benefits to users [12,13], and available guidelines state that HC can be safely used by most women, even perimenopausal or with a family history of breast cancer [47,48,49]. Modern methods are offered in a variety of combinations and forms of administration (oral, injectable, transdermal, subdermal, intrauterine, and intravaginal), aimed to improve the safety profile of contraceptives in terms of side effects and ease of use. This, in turn, allows the use of HC in women with chronic diseases and offers premenopausal women an option for the simultaneous treatment of the symptomatology associated with their reproductive aging. These guidelines also include the necessary contraindications to assist providers and users in making informed decisions, weighing the benefits and risks of each method based on a woman’s preferences and medical conditions. Nevertheless, the increasing incidence of BC in young and premenopausal women has raised the uncertainty about whether the contemporary use of hormonal contraceptives alters BC risks in these users. However, the development of breast cancer responds to multiple factors and not uniquely to the mitogenic activity of female hormones able to induce genomic instability that drives cancer development, as seen in BRCA1 and BRCA2 mutations [1,2,5]. In addition, the evidence is controversial, with some reports showing no association and others showing up to a 40% increased risk of BC for HC users [50]. For these reasons, we conducted this systematic review and meta-analysis of the literature on the risk of BC development in premenopausal women using HC.

The databases PubMed, Cochrane, and EMBASE were consulted. All searches were restricted to articles from 2015 until February 2022, given that we aimed to focus on updated results on all types of modern hormonal contraceptives. Randomized controlled trials were excluded based on scanty evidence, and only peer-reviewed observational studies (prospective cohort studies, case-control studies, and historic cohort studies) were selected. The search strategy identified 724 studies for possible evaluation, of which 60 (8.29%) were selected for full-text evaluation. Finally, 22 (17.74%) studies that meet the inclusion criteria were selected for the quantitative analysis, which included 19 case-control studies, 2 prospective cohort studies, and 1 cross-sectional cohort study. Most of the studies (*n* = 17) were conducted in non-European countries, but 98% of the population included in all reports (*n* = 2,275,304) was included in the European studies (*n* = 5), with the largest population included being in the Danish prospective cohort study (*n* = 1,797,932) [35]. None of the studies represented the Latin American region. 

Regarding the quality criteria, all studies were found to have well-defined and representative cases and controls. However, the method of exposure ascertainment was different and of low quality in around 75% of the studies, as most of them used self-reported questionnaires or unblinded interviews, the hormone formulation of the contraceptive was not discriminated, the exact time of HC use was not based on records, or missing data management of non-respondents was not stated. Only three case-control studies—Joukar et al., Al-Ajmi et al., and Brinton et al. (13.6%% of the 22 included studies)—used tools to avoid recall bias [25,36,44], and two of the cohort studies—Karlsson et al., Mørch et al. (9.09% of the 22 included studies)—used precise instruments to collect clinical information [35,37]. Another difference observed between the studies was the covariates or confounders used by authors to adjust the statistical analysis, which were related to previous evidence of the best-known risk factors for the development of BC. Therefore, this made it possible to analyze the relationship between exposure to HC and the risk of developing BC.

The overall risk analysis including all 22 observational studies showed a statistically significant higher BC risk in every user of HC compared to non-users (pooled OR = 1.33, 95% CI, 1.19 to 1.49; I^2^ = 82%; *p* < 0.01%). A lower pooled overall risk among current users was reported by the Collaborative Group on Hormonal Factors in Breast Cancer in 1996 (RR = 1.24, 95% CI, 1.15 to 1.33; *p* > 0.00001), but the excess risk disappeared 10 or more years after stopping the pill (RR 1.0; 95% CI, 0.96 to 1.05) [51]. Similarly, a lower risk was reported by Kahlenborn et al. among every user of high-dose pills (OR = 1.19; 95% CI, 1.09 to 1.29) [52].

The sensitivity analysis between the groups of cohort and case-control studies included in our analysis showed differences in the pooled OR, which could be explained by the design or sample size of the studies. When exploring the heterogeneity of the group of case-control studies only, this decreases significantly compared with the heterogeneity of the group of cohort studies (I^2^ = 72% vs. I^2^ = 92%; *p* = 0.02). This suggests that study design seems to be a relevant factor influencing the heterogeneity of the results and that cohort studies could have augmented the heterogeneity of the overall results obtained in the first pooled analysis including all 22 reports (I^2^ = 82%). Similarly, the strict definition of menopausal status does not seem to be an important source of heterogeneity (I^2^ = 82%; test of subgroup differences *p* = 0.34).

Considering these analyses, the results of the group of case-control studies seem to provide less heterogeneous and more precise estimates than the results considering all 22 studies. The overall BC risk obtained from this further pooled analysis based on case-control studies only was higher than our first result (pooled OR = 1.44, CI, 1.21 to 1.70, *p* < 0.01) and those reported by other authors such as the following: Romieu et al. (RR = 1.06; 95% CI, 0.98 to 1.14) [53], Delgado-Rodriguez et al. (RR = 1.06; 95% CI, 1.02 to 1.10) [54], Rushton et al. (RR = 1.12; 95% CI, 1.05 to 1.20) [55], and Thomas (RR = 1.0; 95% CI, 1.0 to 1.1) [56] who reported a meta-analysis of case-control studies conducted before 2000. In contrast, the meta-analysis by Kanadys et al. [1], based on studies conducted between 1960 and 2010, found no increased overall risk among all pill users (pooled OR = 1.01; 95% CI, 0.95 to 1.07). In a further analysis of studies conducted between 2000 and 2012, it was also found that BC incidence is significantly increased in users of modern oral contraceptives, with similar results reported by Gierisch et al. [57], Del Pup et al. [58] (RR = 1.08; 95% CI, 1.00 to 1.17), and Anothaisintawee et al. (pooled OR = 1.10; 95% CI, 1.03 to 1.18) [59]. Finally, the risk of developing BC was also high regardless of ethnicity being 1.07 (95% CI: 0.95–1.20) in Caucasian and 1.17 in Asian women (95% CI 0.90 to 0.50), but there were no significative differences between both populations [59]. 

Regarding premenopausal status, our subgroup analysis of the only seven studies that differentiated this population, showed that the overall BC risk is higher for premenopausal women (pooled OR = 1.48, CI, 1.10 to 2.00, I^2^ = 79%; *p* < 0.0001), in comparison to studies that did not make a clear menopausal distinction. The slightly higher overall risk for this population was also reported by Thomas (RR = 1.16; 95% CI, 1.05 to 1.28) [56], Rushton et al. (RR = 1.16; 95% CI, 1.07 to 1.25) [55], Kahlenborn et al. (OR = 1.19; 95% CI, 1.09 to 1.29) [52], and Romieu et al. (RR = 1.17; 95% CI, 0.95 to 1.45) [53]. A higher risk was reported by Delgado-Rodríguez et al. (RR = 1.60; 95% CI, 1.14 to 2.24) [54]. Data from the meta-analysis by Nelson et al.—which included studies and systematic reviews to identify factors that increase the risk of invasive BC specifically in women aged 40–49 years—showed significantly increased risk for current users of oral contraceptives compared to former users or nonusers (RR = 1.30; 95% CI, 1.13 to 1.49), but reduced risk for perimenopausal and postmenopausal women receiving estrogen hormone therapy (RR 0.70; 95% CI, 0.52 to 0.95), while no significant association was found among those receiving combined hormone therapy [60]. Other factors were found to increase the risk of BC in premenopausal women, like having had a second-degree relative with breast cancer, a prior benign breast biopsy, or heterogeneously dense breast tissue at a mammography (RR = between 1.5 to 2.0) as well as nulliparity and being age ≥ 30 at first birth (RR = 1.0 to 1.5). Contrary to the above-mentioned results, Kanadys et al. [1], reported no association in their meta-analysis either in premenopausal users (OR = 1.06; 95% CI, 0.92 to 1.22; *p* = 0.44) or postmenopausal ever-users (OR = 0.99; 95% CI, 0.89 to 1.10). 

Finally, a recent review on the role of sex steroid hormones in female breast development and carcinogenesis is still inconclusive. Bonfiglio et al. concluded that the timing of the onset of hormonal exposure throughout a woman’s lifespan is probably the definitive risk factor for subsequent cancer development, as breast cells are most susceptible to carcinogens before pregnancy until the first full-term pregnancy [61]. Furthermore, the complexity of breast cancer genetics makes it difficult to determine whether women at increased risk experience a selective pattern of responses to exogenic hormones or whether risk accumulates in any of its specific histologic forms, or whether excess risk is determined by age at last exposure, age of onset of use, or duration of exposure.

As can be observed, the interpretation of the evidence regarding the role of HC in the development of BC is difficult due to the heterogeneity among the studies regarding study design, sample size, population studied, and confounding factors. Furthermore, studies generally did not differentiate by route of administration, type of formulation, and time of use of the method. In addition, most of the data sources are questionnaires or interviews, which are affected by recall bias. However, the execution of studies that generate high-quality evidence is not very plausible, given their high cost, the large sample sizes, and the long observation time necessary to differentiate the risk by the type of method to be analyzed.

Another narrative review of prospective and retrospective studies published between 1995 and 2022 [62] shows that the diversity of design and risk factors considered in each study makes it very difficult to analyze the evidence on the strength of the association between HC use and BC. This low quality of evidence on BC risk is outweighed by the reported benefits to women’s health in addition to the effectiveness of any contraception method in preventing pregnancy [63,64]. We, therefore, suggest that contraceptive counseling follows national or WHO recommendations [65] based on selection criteria according to a woman’s health conditions and her reproductive needs, while considering the follow-up of populations at risk of any complication associated with the selected method and applying the recommendations of each country for the early detection of BC.

**Strength and limitations:** As mentioned, this analysis was limited by the heterogeneity of the included studies, the publication biases that favor publications with significant results, and included large studies mostly from European samples with a lack of representation of other regions such as Latin America. The heterogeneity of the studies did not allow for further investigation on the relationship between dose and duration of contraceptive use nor a more detailed analysis of the type of contraceptive used and the risk of breast cancer. Nevertheless, the validity of the results is based on the exhaustive selection of manuscripts, the detailed data extraction, the large sample size of the high-quality studies, and the risk estimates made across study subgroups and populations according to menopausal status.

### Implications for Practice and Research

Based on the results and limitations of this meta-analysis, it is not possible to recommend changes in the well-established practice of contraceptive counseling for individual patients. To reduce the incidence of BC, new users who plan to use a hormonal method for an extended period should agree to use it based on a clinical analysis of their risk factors for BC and the benefits of preventing pregnancy as well as the management of other gynecological conditions that affect their quality of health. Similarly, current users should not be advised to discontinue a method they have been using without side effects, but healthcare providers should take advantage of follow-up visits to take the patient’s anamnesis regarding any changes observed in the breasts. If possible, a medical examination should be carried out for early detection of breast lumps, and further diagnostic tests should be carried out according to clinical findings and personal clinical and genetic risk factors for BC. Changing unhealthy lifestyles should also be encouraged.

The dose and the active ingredient in each contraceptive are other relevant factors that could not be addressed in this study due to heterogeneity in reporting, but they may be important risk factors to consider in future meta-analyses. A next step for future studies in this area could be to investigate the relationship between the duration of use of different drugs used as HCs and the risk of BC in premenopausal women. As most studies on HC have focused on oral contraceptives, it would also be worthwhile to conduct more studies on the relationship between contraceptives other than the pill and BC to learn more about the biological mechanisms of each route of administration on the breast tissue and the risk of BC.

## 5. Conclusions

The results of our meta-analysis of observational studies (cohort and case control) suggest a significantly increased overall risk of BC in users or ever-users of modern hormonal contraceptives (pooled OR = 1.33; CI, 1.19 to 1.49; *p* < 0.01; I^2^ = 82%). The overall risk obtained from the further analysis of case-control studies was higher than our first result (pooled OR = 1.44, CI, 1.21 to 1.70, *p* < 0.01) and was also higher among studies restricted to a strict selection of premenopausal women (pooled OR = 1.48; CI, 1.10 to 2.00; *p* < 0.0001; I^2^ = 79%). As several authors have pointed out, attempting to determine the risk of developing BC among premenopausal women is difficult due to the high heterogeneity observed among the included studies, indicating that our results should be approached with caution when translating them into practice. Since premenopausal women are still fertile and benefit from hormonal contraception for climacteric symptoms, the decision to use should be based on a personalized risk-benefit balance.

Since the studies we gathered are very heterogeneous, there are important factors that could be addressed in future studies. For instance, the effect of different types of oral contraceptives on the risk of breast cancer should be studied in more detail as well as the mechanisms of each route of administration. More careful reporting in medical records of the type, time of use of the chosen method, and beneficial and side effects throughout a woman’s reproductive life could help the quality of future epidemiological analyses to close the evidence gap. 

It is likely that advances in the understanding of the biology of malignant breast tumors and their drivers, together with the development of minimally invasive or nonionizing screening methods, will allow physicians to identify patients at higher risk of developing the disease at an early stage and thus recommend non-hormonal methods of preventing pregnancy. Future drugs that are more selective in their pharmacological action on breast cells could also increase the safety of new hormonal methods.

## Figures and Tables

**Figure 1 cancers-15-05624-f001:**
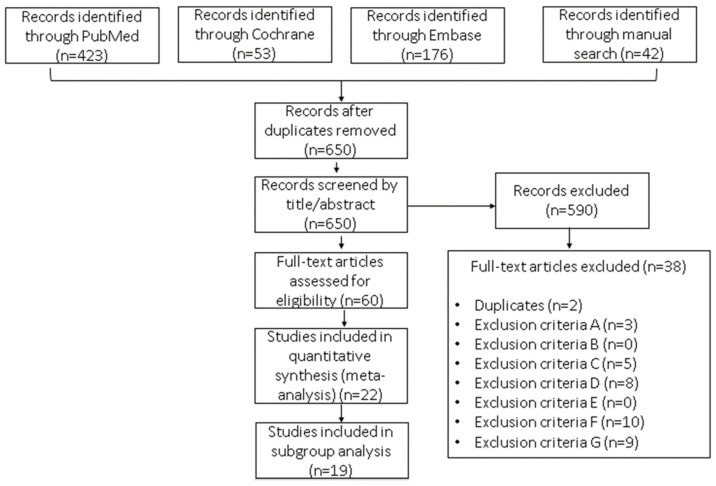
Flow diagram of the selection process for studies included in this meta-analysis: hormonal contraceptive use as a potential risk factor for breast cancer. The reasons for exclusion were as follows: (A) not the intended outcome—e.g., the study outcome was mortality rate but not risk of BC; (B) not the intended exposure—e.g., hormone replacement therapy; (C) not the intended participant—e.g., males, cancer survivors, women with a family history of BC, or underage or post-menopausal women; (D) not the intended study design or not an observational study; (E) not an original research study—e.g., book chapters, abstracts, or other meta-analyses; (F) not quantitative data reported—e.g., odds ratios, risk ratios, or hazard ratios; and (G) studies focusing exclusively on BC subtypes.

**Figure 2 cancers-15-05624-f002:**
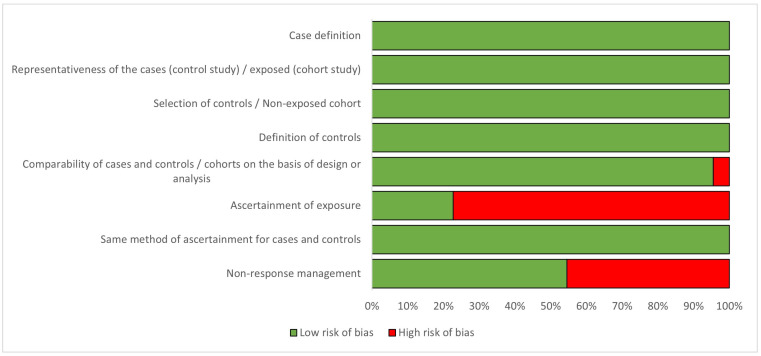
Percentage of risk of bias for studies included in the analysis, according to the Newcastle-Ottawa Quality Assessment Scale [20].

**Figure 3 cancers-15-05624-f003:**
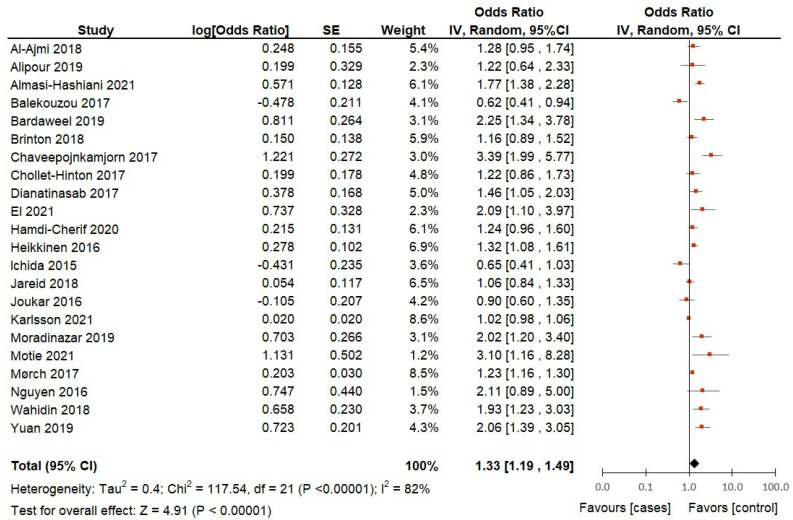
Hormonal contraceptive use and the risk of breast cancer in adult women according to case-control and cohort studies included in the analysis.

**Figure 4 cancers-15-05624-f004:**
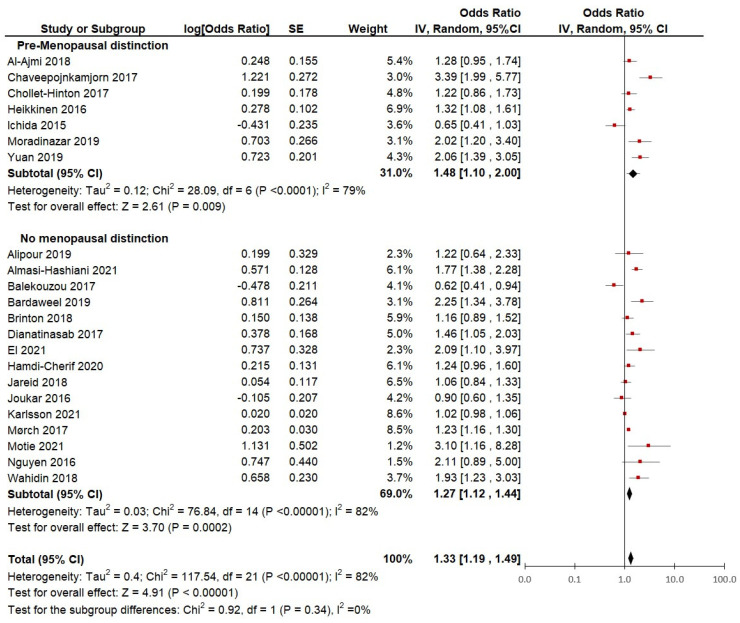
Subgroup analysis of hormonal contraceptive use and the risk of breast cancer between groups of case-control and cohort studies distinguished by menopausal status.

**Figure 5 cancers-15-05624-f005:**
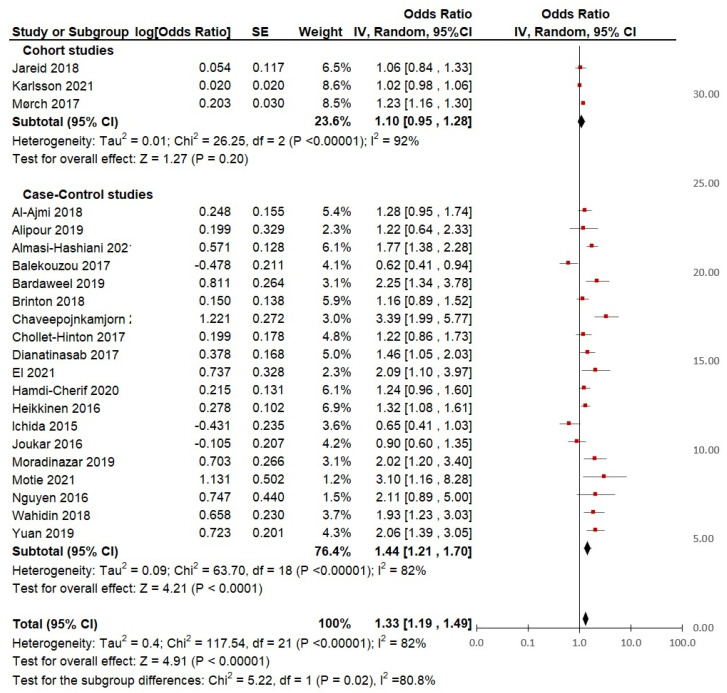
Subgroup analysis of hormonal contraceptive use and the risk of breast cancer between groups of cohort and case-control studies.

**Table 2 cancers-15-05624-t002:** Confounders and covariates considered in studies included in the analysis.

Author, Year of Publication	Confounders and Covariates
Heikkinen et al., 2016 [33]	HC use, use of HR IUD, and HRT, age at menarche, parity, family history of BC, BMI, education, smoking, and alcohol use.
Ichida et al., 2015 [38]	Age, parity, breastfeeding, family history of BC.
Joukar et al., 2016 [25]	Alcohol consumption, family history OF BC, nutrition, history of benign breast tumors, X-ray on the chest.
Nguyen et al., 2016 [39]	Age at menarche, parity, breastfeeding time, BMI.
Balekouzou et al., 2017 [41]	Age, occupation, education level, residence, marital status, age at menarche, menstrual cycles, term pregnancy, abortion, parity, breastfeeding, No. of births.
Mørch et al., 2017 [35]	Level of education, parity, polycystic ovary syndrome, endometriosis, family history of breast or ovarian cancer, body-mass index, smoking status, and age of the woman at first delivery for parous women.
Dianatinasab et al., 2017 [26]	Age, education, occupation, family history, smoking, hysterectomy, hair coloring, physical exercise, X-rays, BMI, occupation, regular bedtime.
Jareid et al., 2018 [34]	Age at the start of follow-up, BMI, physical activity level, maternal history of breast cancer, OC use, and menopausal status at the start of the follow-up.
Chaveepojnkamjorn et al., 2017 [45]	Family history of BC, history of benign breast tumor, younger age at menarche, parity, miscarriage, BMI, passive smoking, and multivitamin use.
Chollet-Hinton et al., 2017 [43]	All models controlled for study, diagnosis year, geographic region, and education status to account for differences between studies. Likewise: age, study site, index year, geographic location, education level, and confounders, by model: BMI, parity; age at first live birth; age at last live birth, breastfeeding duration, oral contraceptive use/duration/residency.
Wahidin et al., 2018 [46]	Oral contraceptive use, age, unhealthy diet, history of benign tumor, breastfeeding status, and hospital.
Al-Ajmi et al., 2018 [36]	Age, family history of BC in first-degree relatives, and deprivation score.
Brinton et al., 2018 [44]	Age, race, a combined variable of the number of births and age at first childbirth, history of a mammogram, and menopausal status.
Yuan et al., 2019 [40]	Age, per capita annual income, history of active and passive smoking, alcohol drinking, and history of live birth.
Alipour et al., 2019 [27]	Age and place of residency, parity, BMI, family history of any cancer, duration of OC.
Moradinazar et al., 2019 [28]	Demographic variables and body mass index.
Bardaweel et al., 2019 [29]	Previous pregnancy, menopausal status, personal history of cancers other than BC, family history of cancer; first or second-degree relatives, smoking status, use of HRT, number of previous miscarriages, age at puberty and menopause, use of OCs and duration of use.
Hamdi-Cherif et al., 2020 [42]	Age in quinquennia and education.
Almasi-Hashiani et al., 2021 [30]	Parity, menopausal status, age at first pregnancy, age at first marriage, breastfeeding duration, OC use) and anthropometric (age, height, weight, body mass index) and socioeconomic factors as potential confounders. (Educational level, occupation, marital status.
Motie et al., 2021 [31]	Age, residency, ethnicity, occupation, height and weight, BMI, marital status, contraception method, number of children, number of pregnancies and abortions, breastfeeding duration, age at the first full-term pregnancy, age, at menarche, and menopause. History of previous diseases or radiation therapy, previous mammography, family history of cancer, alcohol consumption, tobacco usage or addiction, and drug history.
El Sharif et al., 2021 [32]	Age at menarche, ever OC use for ≥2 months, use of HRT, age at first marriage, parity, age at first pregnancy, age at first delivery, number of full-term pregnancies, ever breastfeeding, age at first breastfeeding, total breastfeeding.
Karlsson et al., 2021 [37]	Age, body mass index, Townsend deprivation index (TDI), year of birth, smoking status, age at menarche, hormone replacement treatment (HRT) use, number of live births, as well as menopausal and hysterectomy status as covariates.

BC = Breast cancer; BMI = Body mass index; HR IUD = Hormone-releasing intrauterine device; HRT = Hormone replacement therapy; IA = Induced abortion; OC = Oral contraceptive pill.

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
