# Peer review of "Hormonal Contraception and the Risk of Breast Cancer in Women of Reproductive Age: A Meta-Analysis"

_cancers, 2023, doi:10.3390/cancers15235624_

Round 1
Reviewer 1 Report
Comments and Suggestions for Authors
The authors studied in this review manuscript the Hormonal contraception and risk of breast cancer in women of reproductive age. it is big work and give excellent information to the readers and is possess high interesting object for researcher.
In this review all details explained in good manner.
Author Response
Dear Reviewer,
Your words are an honour to us and an encouragement to move forward.
A new version of our manuscript was uploaded giving the information required for the other reviewers.
Kind reagrds
Reviewer 2 Report
Comments and Suggestions for Authors
Line 37 Please correct: The I² statistic (I-squared statistic)
Line 42: BC and HC is not described before in the abstract text.
Citations in text is not according MDPI instruction for reference format.
Line 73 is it MEC or ECM?
Line 134: the strategy in Appendix A is just for PubMed. Did you use the same for EMBASE and Cochrane?
Line 191: "081" change to "0.81"
Lines 200 and 201: both "2"s are superscript.
Abbreviation in Figure 1 needs to be explained.
Table 2 needs heading for columns.
Comments on the Quality of English LanguageLines 41 and 42: "higher risk" than what?
"RevMan" in all text needs to be written similar.
Author Response
Dear Reviewer,
thanks for your comments that helped us to improve the quality of our manuscript.
- Several typing errors:
- Line 37 Please correct: The I² statistic (I-squared statistic)
- Line 42: BC and HC is not described before in the abstract text.
- Citations in text is not according MDPI instruction for reference format. / RevMan" in all text needs to be written similar.
- Line 73 is it MEC or ECM?
- Line 191: "081" change to "0.81"
- Lines 200 and 201: both "2"s are superscript.
- Abbreviation in Figure 1 needs to be explained.
- Table 2 needs heading for columns.
- Lines 41 and 42: "higher risk" than what?
Response: The aforementioned errors were corrected according to.
______________________________________________________
- Line 134: the strategy in Appendix A is just for PubMed. Did you use the same for EMBASE and Cochrane?
Response: No, the search strategies from EMBASE and Cochrane have been added to Appendixes.
Reviewer 3 Report
Comments and Suggestions for Authors
This manuscript entitled "Hormonal contraception and risk of breast cancer in women of reproductive age: a meta-analysis" described a meta-analysis of the literature on lifetime use of hormonal contraceptives and breast cancer risk from observational studies in women of reproductive age. The following issues should be addressed to further improve the manuscript.
1. Authors are advised to mention the specific objectives of this work clearly in the Introduction section.
2. The language should be polished a little bit. There are several instances where the language could be improved for clarity and precision.
3. The authors should re-read the full manuscript and reorganize chronologically the abbreviations and correct typos.
4. In the manuscript, the following references may be considered:
DOI: 10.2147/BCTT.S390664
DOI: 10.3390/ijms22094597
5. In conclusion, can the authors add a few lines about the future clinical directions to reduce breast cancer risk in hormonal contraceptives.
Comments on the Quality of English Language-
Author Response
Dear Reviewer,
thanks for your comments taht helped us to improve the quality of our manuscript:
- Authors are advised to mention the specific objectives of this work clearly in the Introduction section.
Response: a new paragraph at the end of the introduction to clarify our research aims in more detail was added, as follows:
Given the aforementioned background, the main aim of this study was to investigate the risk of lifetime use of HC in the development of BC in women of reproductive age. To this end, we aimed to summarize the existing evidence regarding the relationship between HC use and BC as reported in peer-reviewed observational studies conducted with women of reproductive age by performing a systematic review of the literature, with a meta-analysis. In addition, we aimed to investigate if differences by study design, report of menopausal status, and quality of the studies had an influence in the heterogeneity of the results.
______________________________________________________
- The language should be polished a little bit. There are several instances where the language could be improved for clarity and precision.
Response: English spelling corrections were made along the text.
______________________________________________________
- The authors should re-read the full manuscript and reorganize chronologically the abbreviations and correct typos.
Response: Abbreviations and typos errors were reorganized and corrected along the text.
______________________________________________________
- In the manuscript, the following references may be considered: DOI: 10.2147/BCTT.S390664 and DOI: 10.3390/ijms22094597.
Response: the suggested reference were not considered for the analysis because do not match the topic (DOI: 10.3390/ijms22094597) or the selection criteria for the present metaanalysis, thus was commented in the discussion (DOI: 10.2147/BCTT.S390664), as follows:
Another narrative review of prospective and retrospective studies published between 1995 and 2022 shows that the diversity of design and risk factors considered in each study makes it very difficult to analyze the evidence on the strength of the association between HC use and BC.
______________________________________________________
In conclusion, can the authors add a few lines about the future clinical directions to reduce breast cancer risk in hormonal contraceptives.
Response: The following sentences was added: to the conclusion to address this suggestion:
It is likely that advances in the understanding of the biology of malignant breast tumors and their drivers, together with the development of minimally invasive or nonionizing screening methods, will allow physicians to identify patients at higher risk of developing the disease at an early stage and thus recommend non-hormonal methods of preventing pregnancy. Future drugs that are more selective in their pharmacological action on breast cells could also increase the safety of new hormonal methods.
Reviewer 4 Report
Comments and Suggestions for Authors
This is an important study, which is of great potential interest to a wide audience of readers. To my opinion, the manuscript is too descriptive and too focused on formal procedures and data, instead of delivering a clear medical or biological message.
Abstract: lines 33-41 provide unnecessary information.
Introduction: I suggest to briefly describe mechanisms of action of oral contraceptives, provide an overview on the main categories of these drugs and explain why their use may be associated with breast cancer risk.
The description of studies: please consider the types of contraceptives.
Table 2 provides the list of factors influencing BC risk. How the use of (different types of) oral contraceptives affects the breast cancer probability for women at risk, e.g., females with cancer family history, BRCA1/2 mutations etc.? Is there any category of women for whom the use of hormonal contraceptives indeed looks risky, or, vice versa, does not render a harm?
Please consider whether some table/figures can be moved to Supplements and whether the description of the data and discussion can be made more concise.
How to proceed with the results of the study? What are the recommendations? How to design future studies in this field?
Author Response
Dear Reviewer,
thanks for your comments that helped us to improve the quality of our manuscript:
This is an important study, which is of great potential interest to a wide audience of readers. To my opinion, the manuscript is too descriptive and too focused on formal procedures and data, instead of delivering a clear medical or biological message.
- Abstract: lines 33-41 provide unnecessary information.
Response: Since this part of the abstract describes important information about the methods, we have kept some relevant information but have reduced the length of this section and have removed information from the methods that may not be relevant for the abstract, namely the description of the quality assessment and risk of bias assessment.
______________________________________________________
- Introduction: I suggest to briefly describe mechanisms of action of oral contraceptives, provide an overview on the main categories of these drugs and explain why their use may be associated with breast cancer risk.
Response: the following paragraph was added:
Regarding synthetic estrogens and progestins, used for contraception or hormone replacement therapy during menopause, epidemiological studies and clinical trials show conflicting results regarding the risk of developing BC. These inconsistencies could be the result of multiple factors, such as the design of the studies, the confounding factors considered in each study, the non-comparability of the data from the reviews, the possibility of an undetected malignancy at the time of initiation of hormonal contraception (HC), or due to the heterogeneity of the biological effect of steroid hormones on breast cells in the presence of other risk factors for BC. According to different meta-analyses of case-control studies, the use of modern oral contraceptives has different effects on the risk of BC subtypes. In 2017 Li et al. [8], reported a statistically significant higher risk of triple-negative breast cancer (TNBC) among users (OR = 1.31, 95% CI = 1.18-1.45) in comparison to healthy population (OR = 1.21, 95% CI = 1.01-1.46). Similar results were found in two meta-analyses published recently. Barańska et al. [9], reported an increased risk for TNBC (OR = 1.37; 95% CI: 1.13 to 1.67) and negative-estrogen receptor (ER-) cancers (OR = 1.20; 95% CI: 1.03 to 1.40), but reduced risk for positive-estrogen receptor (ER+) tumors (OR = O.92; 95% CI: 0.86 to 0.99). Mao X et al. [10], show that prolonged oral contraceptive use is associated with a 16% higher risk of TNBC (RR: 1.16; 95% CI: 1.05 to 1.29), but there are no statistically significant associations for luminal A, luminal B or HER2 tumors. The evidence is scant since just a few prospective controlled studies have been conducted in the past decade.
Modern low-dose HC methods, which are available in a variety of hormonal combinations and forms of administration (oral, injectable, transdermal, subdermal, intrauterine, and intravaginal), have proved to be effective in preventing pregnancy in healthy young and premenopausal women. However, the reported risk of BC development is still one of the most frequent reasons given for the non-acceptance of HC use [11], even though the long-term benefits of HC are thought to be multiple and greater than this potential risk [12, 13].
______________________________________________________
- The description of studies: please consider the types of contraceptives.
Response: Information has been added to Table 1, as presented in the original paper, on the type of contraceptives analyzed in the study.
______________________________________________________
- Table 2 provides the list of factors influencing BC risk. How the use of (different types of) oral contraceptives affects the breast cancer probability for women at risk, e.g., females with cancer family history, BRCA1/2 mutations etc.? Is there any category of women for whom the use of hormonal contraceptives indeed looks risky, or, vice versa, does not render a harm?
Response: This is an interesting question, however due to the heterogeneity of the studies, there were methodological limitations that did not allow us to conduct analysis in which we could find the relationship between specific types of contraceptives and specific-subtype BC cancer risk. Most of the studies found used hormonal oral contraceptives as the exposure variable, however the specific hormones were rarely mentioned. Other studies investigated overall use of hormonal contraception but did not distinguish between types. Regarding the category of woman, the heterogeneity of the studies also led to limitations that did not allow us to investigate more granular variables further (for example, dose and length of use of contraceptives) but we tried to control for this heterogeneity by performing subgroup analysis using papers with a strict description of pre-menopausal status. We have added a clarification on this in the limitations.
______________________________________________________
- Please consider whether some table/figures can be moved to Supplements and whether the description of the data and discussion can be made more concise.
Response: We have moved two figures to the appendices, namely, the risk of bias evaluation per paper, and the funnel plot.
______________________________________________________
- How to proceed with the results of the study? What are the recommendations? How to design future studies in this field?
Response: We have added an extra paragraph at the end of the manuscript with some directions for future studies, as follows:
Implications for practice and research. Based on the results and limitations of this meta-analysis, it is not possible to recommend changes in the well-established practice of contraceptive counseling for individual patients. To reduce the incidence of BC, new users who plan to use a hormonal method for an extended period should agree to use it based on a clinical analysis of their risk factors for BC and the benefits of preventing pregnancy, as well as the management of other gynecological conditions that affect their quality of health. Similarly, current users should not be advised to discontinue a method they have been using without side effects, but healthcare providers should take advantage of follow-up visits to take the patient's anamnesis regarding any changes observed in her breasts. If possible, a medical examination should be carried out for early detection of breast lumps, and further diagnostic tests should be carried out according to clinical findings and personal clinical and genetic risk factors for BC. Changing unhealthy lifestyles should also be encouraged.
The dose and the active ingredient in each contraceptive are other relevant factors that could not be addressed in this study due to heterogeneity in reporting, but which may be important risk factors to consider in future meta-analyses. A next step for future studies in this area could be to investigate the relationship between the duration of use of different drugs used as HCs and the risk of BC in premenopausal women. As most studies on HC have focused on oral contraceptives, it would also be worthwhile to conduct more studies on the relationship between contraceptives other than the pill and BC, to learn more about the biological mechanisms of each route of administration on the breast tissue and the risk of BC.
CONCLUSION, last paragraph: Since the studies we gathered are very heterogeneous, there are important factors that could be addressed in future studies, for instance, the effect of different types of oral contraceptives on the risk of breast cancer should be studied in more detail, as well as the mechanisms of each route of administration. More careful reporting in medical records of the type, time of use of the chosen method, and beneficial and side effects throughout the woman's reproductive life could help the quality of future epidemiological analyses to close the evidence gap.
It is likely that advances in the understanding of the biology of malignant breast tumors and their drivers, together with the development of minimally invasive or nonionizing screening methods, will allow physicians to identify patients at higher risk of developing the disease at an early stage and thus recommend non-hormonal methods of preventing pregnancy. Future drugs that are more selective in their pharmacological action on breast cells could also increase the safety of new hormonal methods.
Round 2
Reviewer 3 Report
Comments and Suggestions for Authors
The authors addressed my concerns.
Reviewer 4 Report
Comments and Suggestions for Authors
The authors have addressed all issues raised